# Tiny Graph Convolutional Networks with Topologically Consistent Magnitude Pruning

**Hichem Sahbi**
Sorbonne University, UPMC, CNRS, LIP6, France.
`hichem.sahbi@sorbonne-universite.fr`

## Abstract

Magnitude pruning is one of the mainstream methods in lightweight architecture design whose goal is to extract subnetworks with the largest weight connections. This method is known to be successful, but under very high pruning regimes, it suffers from topological inconsistency which renders the extracted subnetworks disconnected, and this hinders their generalization ability. In this paper, we devise a novel end-to-end Topologically Consistent Magnitude Pruning (TCMP) method that allows extracting subnetworks while guaranteeing their topological consistency. The latter ensures that only accessible and co-accessible — impactful — connections are kept in the resulting lightweight architectures. Our solution is based on a novel reparametrization and two supervisory bi-directional networks which implement accessibility/co-accessibility and guarantee that only connected subnetworks will be selected during training. This solution allows enhancing generalization significantly, under very high pruning regimes, as corroborated through extensive experiments, involving graph convolutional networks, on the challenging task of skeleton-based action recognition.

## 1 Introduction

Deep convolutional networks are nowadays becoming mainstream in solving many pattern classification tasks including visual recognition [22, 4, 3, 6]. Their principle consists in training convolutional filters together with pooling and attention mechanisms that maximize classification performances. Many existing convolutional networks were initially dedicated to grid-like data, including images [23, 25, 24, 27]. However, data sitting on top of irregular domains (such as skeleton graphs in action recognition) require extending convolutional networks to general graph structures, and these extensions are known as graph convolutional networks (GCNs) [9, 26]. Two families of GCNs exist in the literature: spectral and spatial. Spectral methods are based on graph Fourier transform [42, 31–34, 29, 30, 11, 12] while spatial ones rely on message passing and attention [36–41]. Whilst spatial GCNs have been relatively more effective compared to spectral ones, their precision is reliant on the attention matrices that capture context and node-to-node relationships [43]. With multi-head attention, GCNs are more accurate but overparametrized and computationally overwhelming.

Many solutions are proposed in the literature to reduce time and memory footprint of convolutional networks including GCNs [45–50]. Some of them pretrain oversized networks prior to reduce their computational complexity (using distillation [51–59], linear algebra [67], quantization [63] and pruning [60–62]), whilst others build efficient networks from scratch using neural architecture search [68]. In particular, pruning methods, either unstructured or structured are currently mainstream, and their principle consists in removing connections whose impact on the classification performance is the least noticeable. Unstructured pruning [62, 63] proceeds by dropping out connections individually using different proxy criteria, such as weight magnitude, and then retraining the resulting pruned networks. In contrast, structured pruning [64, 66] removes groups of connections, entire filters or subnetworks using different mechanisms such as grouped sparsity. However, existing pruning methods either

37th Conference on Neural Information Processing Systems (NeurIPS 2023 / WANT).

structured or unstructured suffer from several drawbacks. On the one hand, structured pruning may reach high speedup on usual hardware, but its downside resides in the rigidity of the class of learnable architectures. On the other hand, unstructured pruning is more flexible, but its discrimination is limited at high pruning regimes due to *topological disconnections*, and handling the latter is highly intractable as adding or removing any connection *combinatorially* affects the others.

As contemporary network sizes grow into billions of parameters, studying high compression regimes has been increasingly important on very large network architectures. Nevertheless, pruning mid-size (but still heavy) architectures, including GCNs, is even more challenging as this usually leads to highly disconnected and untrainable subnetworks, even at reasonably (not very) large pruning rates. Hence, we target our contribution towards mid-size network architectures including GCNs in order to fit not only the usual edge devices, such as smartphones, but also highly *miniaturized* devices endowed with very limited computational resources (e.g., smart glasses). Considering the aforementioned issues, our contribution in this paper includes a new lightweight design which guarantees the topological consistency of the extracted subnetworks. Our proposed solution is variational and proceeds by training pruning masks and weight parameters that maximize classification performances while guaranteeing the *accessibility* of the unpruned connections (i.e., their reachability from the network input) and their *co-accessibility* (i.e., their actual contribution in the evaluation of the output). Hence, only topologically consistent (accessible and co-accessible) subnetwork connections are combinatorially selected. Extensive experiments, on the challenging task of skeleton-based action recognition, show the outperformance of our proposed TCMP method.

## 2 A Glimpse on GCNs

Let $\mathcal{S} = \{\mathcal{G}_i = (\mathcal{V}_i, \mathcal{E}_i)\}_i$ denote a collection of graphs with $\mathcal{V}_i$, $\mathcal{E}_i$ being respectively the nodes and the edges of $\mathcal{G}_i$. Each graph $\mathcal{G}_i$ (denoted for short as $\mathcal{G} = (\mathcal{V}, \mathcal{E})$) is endowed with a signal $\{\varphi(u) \in \mathbb{R}^s : u \in \mathcal{V}\}$ and associated with an adjacency matrix $\mathbf{A}$ with each entry $\mathbf{A}_{uu'} > 0$ iff $(u, u') \in \mathcal{E}$ and 0 otherwise. GCNs aim at learning a set of $C$ filters $\mathcal{F}$ that define convolution on $n$ nodes of $\mathcal{G}$ (with $n = |\mathcal{V}|$) as $(\mathcal{G} \star \mathcal{F})_\mathcal{V} = f(\mathbf{A} \, \mathbf{U}^\top \, \mathbf{W})$, here $^\top$ stands for transpose, $\mathbf{U} \in \mathbb{R}^{s \times n}$ is the graph signal, $\mathbf{W} \in \mathbb{R}^{s \times C}$ is the matrix of convolutional parameters corresponding to the $C$ filters and $f(.)$ is a nonlinear activation applied entrywise. In $(\mathcal{G} \star \mathcal{F})_\mathcal{V}$, the input signal $\mathbf{U}$ is projected using $\mathbf{A}$ and this provides for each node $u$, the aggregate set of its neighbors. Entries of $\mathbf{A}$ could be handcrafted or learned so $(\mathcal{G} \star \mathcal{F})_\mathcal{V}$ implements a convolutional block with two layers; the first one aggregates signals in $\mathcal{N}(\mathcal{V})$ (sets of node neighbors) by multiplying $\mathbf{U}$ with $\mathbf{A}$ while the second layer achieves convolution by multiplying the resulting aggregates with the $C$ filters in $\mathbf{W}$. Learning multiple adjacency (also referred to as attention) matrices (denoted as $\{\mathbf{A}^k\}_{k=1}^K$) allows us to capture different contexts and graph topologies when achieving aggregation and convolution. With multiple matrices $\{\mathbf{A}^k\}_k$ (and associated convolutional filter parameters $\{\mathbf{W}^k\}_k$), $(\mathcal{G} \star \mathcal{F})_\mathcal{V}$ is updated as $f\left(\sum_{k=1}^K \mathbf{A}^k \mathbf{U}^\top \mathbf{W}^k\right)$. Stacking aggregation and convolutional layers, with multiple matrices $\{\mathbf{A}^k\}_k$, makes GCNs accurate but heavy. We propose subsequently a method that makes our network architectures lightweight and still effective.

## 3 Magnitude Pruning

In the rest of this paper, a given GCN is subsumed as a multi-layered neural network $g_\theta$ whose weights defined as $\theta = \{\mathbf{W}^1, \ldots, \mathbf{W}^L\}$, with $L$ being its depth, $\mathbf{W}^\ell \in \mathbb{R}^{d_{\ell-1} \times d_\ell}$ its $\ell^{\text{th}}$ layer weight tensor, and $d_\ell$ the dimension of $\ell$. The output of a given layer $\ell$ is defined as $\phi^\ell = f_\ell(\mathbf{W}^{\ell\top} \phi^{\ell-1})$, $\ell \in \{2, \ldots, L\}$, being $f_\ell$ an activation function. Without a loss of generality, we omit the bias in the definition of $\phi^\ell$. Magnitude Pruning (MP) consists in zeroing the smallest weights in $g_\theta$ (up to a pruning rate), while retraining the remaining weights. A relaxed variant of MP is obtained by multiplying $\mathbf{W}^\ell$ with a differentiable mask $\psi(\mathbf{W}^\ell)$ applied entrywise to $\mathbf{W}^\ell$. The entries of $\psi(\mathbf{W}^\ell)$ are set depending on whether the underlying layer connections are kept or removed, so $\phi^\ell = f_\ell((\mathbf{W}^\ell \odot \psi(\mathbf{W}^\ell))^\top \phi^{\ell-1})$, here $\odot$ stands for the element-wise matrix product. In this definition, $\psi(\mathbf{W}^\ell)$ enforces the prior that smallest weights should be removed from the network. In order to achieve magnitude pruning, $\psi$ must be symmetric, bounded in $[0, 1]$, and $\psi(\omega) \rightsquigarrow 1$ when $|\omega|$ is sufficiently large and $\psi(\omega) \rightsquigarrow 0$ otherwise[1].

---

[1] A possible choice, used in practice, that satisfies these four conditions is $\psi(\omega) = 2\sigma(\omega^2) - 1$ with $\sigma$ being the sigmoid function.

Pruning is achieved using a global loss as a combination of a cross entropy term denoted as $\mathcal{L}_e$, and a budget cost which measures the difference between the targeted cost (denoted as $c$) and the actual number of unpruned connections

$$\min_{\{\mathbf{W}^\ell\}_\ell} \mathcal{L}_e\big(\{\mathbf{W}^\ell \odot \psi(\mathbf{W}^\ell)\}_\ell\big) + \lambda\big(\sum_{\ell=1}^{L-1} \mathbf{1}_{d_\ell}^\top \psi(\mathbf{W}^\ell)\mathbf{1}_{d_{\ell+1}} - c\big)^2, \tag{1}$$

here $\mathbf{1}_{d_\ell}$ is a vector of $d_\ell$ ones. Eq. 1 focuses on minimizing the budget loss (with $\lambda$ sufficiently large) while progressively making $\{\psi(\mathbf{W}^\ell)\}_\ell$ crisp (almost binary) by linearly annealing the temperature of the sigmoid function that defines $\psi$. As training evolves, the right-hand side term reaches its minimum and stabilizes while the gradient of the global loss becomes dominated by the gradient of the left-hand side term, and this maximizes further the classification performances.

## 4 Proposed Method: TCMP

The aforementioned pruning formulation is relatively effective (as shown later in experiments), however, it suffers from several drawbacks. On the one hand, removing connections independently may result into *topologically inconsistent* network architectures (see section 4.1), i.e., either completely disconnected or having isolated connections. On the other hand, high pruning rates may lead to an over-regularization effect and hence weakly discriminant lightweight networks, especially when the latter include isolated connections (see again later experiments). In what follows, we introduce a more principled pruning framework that guarantees the topological consistency of the pruned networks and allows improving generalization even at very high pruning rates.

### 4.1 Accessibility and Co-accessibility

Our formal definition of topological consistency relies on two principles: *accessibility and co-accessibility* of connections in $g_\theta$. Let's remind $\psi(\mathbf{W}_{ij}^\ell)$ as a crisp (binary) function that indicates the presence or absence of a connection between the i-th and the j-th neurons of layer $\ell$. This connection is referred to as accessible if $\exists i_1, \ldots, i_{\ell-1}$, s.t. $\psi(\mathbf{W}_{i_1,i_2}^1) = \cdots = \psi(\mathbf{W}_{i_{\ell-1},i}^{\ell-1}) = 1$, and it is co-accessible if $\exists i_{\ell+1}, \ldots, i_L$, s.t. $\psi(\mathbf{W}_{j,i_{\ell+1}}^{\ell+1}) = \cdots = \psi(\mathbf{W}_{i_{L-1},i_L}^L) = 1$.

Considering $\mathbf{S}_a^\ell = \psi(\mathbf{W}^1)\,\psi(\mathbf{W}^2)\ldots\psi(\mathbf{W}^{\ell-1})$ and $\mathbf{S}_c^\ell = \psi(\mathbf{W}^{\ell+1})\,\psi(\mathbf{W}^{\ell+2})\ldots\psi(\mathbf{W}^L)$, and following the above definition, it is easy to see that a connection between $i$ and $j$ is accessible (resp. co-accessible) iff the i-th column (resp. j-th row) of $\mathbf{S}_a^\ell$ (resp. $\mathbf{S}_c^\ell$) is different from the null vector. A network architecture is called topologically consistent iff all its connections are both accessible and co-accessible. Accessibility guarantees that incoming connections to the i-th neuron carry out effective activations resulting from the evaluation of $g_\theta$ up to layer $\ell$. Co-accessibility is equivalently important and guarantees that the outgoing activation from the j-th neuron actually contributes in the evaluation of the network output. A connection — not satisfying accessibility or co-accessibility and even when its magnitude is large — becomes useless and should be removed when $g_\theta$ is pruned.

For any given network architecture, parsing all its topologically consistent subnetworks and keeping only the one that minimizes Eq. 1 is highly combinatorial. Indeed, the accessibility of a given connection depends on whether its preceding and subsequent ones are kept or removed, and any masked connections may affect the accessibility of the others. A heuristic is proposed in [35] as a greedy approach to prune networks while guaranteeing their topological consistency; however, this approach is clearly suboptimal as (i) topologically consistent subnetwork selection is *decoupled* from (ii) weight retraining. In what follows, we introduce our main contribution (TCMP) that *couples* both steps (i) and (ii) during network pruning using two supervisory accessibility networks.

### 4.2 Accessibility and Co-Accessibility Networks

Our solution relies on two supervisory networks that measure accessibility and co-accessibility of connections in $g_\theta$. These two networks, denoted as $\phi_r$ and $\phi_l$, have exactly the same architecture as $g_\theta$ with only a few differences: indeed, $\phi_r$ measures accessibility and inherits the same connections in $g_\theta$ with the only difference that their weights correspond to $\{\psi(\mathbf{W}^\ell)\}_\ell$ instead of $\{\mathbf{W}^\ell \odot \psi(\mathbf{W}^\ell)\}_\ell$. Similarly, $\phi_l$ inherits the same connections and weights as $\phi_r$, however these connections are reversed in order to measure accessibility in the opposite direction (i.e., co-accessibility). Note that weights $\{\mathbf{W}^\ell\}_\ell$ are shared across all the networks $g_\theta$, $\phi_r$ and $\phi_l$.

Considering the definition of accessibility and co-accessibility, one may define layerwise outputs $\phi_r^\ell := h\big((\phi_r^{\ell-1}\phi_l^{\ell^\top} \odot \psi(\mathbf{W}_{\ell-1}))^\top \phi_r^{\ell-1}\big)$, and $\phi_l^\ell := h\big((\phi_r^\ell\phi_l^{\ell+1^\top} \odot \psi(\mathbf{W}_\ell))\phi_l^{\ell+1}\big)$ being $\phi_r^1 = \mathbf{1}_{d_1}$, $\phi_l^L = \mathbf{1}_{d_L}$, $\mathbf{1}_{d_1}$ the vector of $d_1$ ones and $h$ the Heaviside activation. With $\phi_r^\ell$ and $\phi_l^\ell$, the non-zero entries of the matrix $(\phi_r^\ell\phi_l^{\ell+1^\top}) \odot \psi(\mathbf{W}^\ell)$ correspond to selected connections in $g_\theta$ which are also accessible and co-accessible. By plugging this matrix into Eq. 1, we redefine our topologically consistent pruning loss

$$\mathcal{L}_e\big(\{\mathbf{W}^\ell \odot \psi(\mathbf{W}^\ell) \odot \phi_r^\ell\phi_l^{\ell+1^\top}\}_\ell\big) + \lambda\big(\sum_{\ell=1}^{L-1} \phi_r^{\ell^\top}\psi(\mathbf{W}^\ell)\phi_l^{\ell+1} - c\big)^2. \tag{2}$$

It is clear that accessibility networks in Eqs. 2 are interdependent and cannot be modeled using standard feedforward networks, so more complex (highly recursive and interdependent) networks should be considered which also lead to exploding gradient. In order to make accessibility and co-accessibility networks in Eqs. 2 simpler and still trainable with standard feedforward networks, we constrain entries of $\psi(\mathbf{W}_\ell)$ to take non-zero values *iff* the underlying connections are kept and accessible/co-accessible; in other words, $\phi_r^{\ell^\top}\psi(\mathbf{W}_\ell)\phi_l^{\ell+1}$ should approximate $\mathbf{1}_{d_\ell}^\top\psi(\mathbf{W}_\ell)\mathbf{1}_{d_{\ell+1}}$ in order to guarantee that (i) unpruned connections are necessarily accessible/co-accessible and (ii) non accessible ones are necessarily pruned. Hence, instead of Eqs. 2, a surrogate loss is defined as

$$\mathcal{L}_e\big(\{\mathbf{W}^\ell \odot \psi(\mathbf{W}^\ell) \odot \phi_r^\ell\phi_l^{\ell+1^\top}\}_\ell\big) + \lambda\big(\sum_{\ell=1}^{L-1} \phi_r^{\ell^\top}\psi(\mathbf{W}^\ell)\phi_l^{\ell+1} - c\big)^2$$
$$+\eta\sum_{\ell=1}^{L-1}\big[\mathbf{1}_{d_\ell}^\top\psi(\mathbf{W}_\ell)\mathbf{1}_{d_{\ell+1}} - \phi_\ell^{r^\top}\psi(\mathbf{W}_\ell)\phi_{\ell+1}^l\big], \tag{3}$$

with now $\phi_r^\ell := h\big(\psi(\mathbf{W}_{\ell-1})^\top \phi_r^{\ell-1}\big)$ and $\phi_l^\ell := h\big(\psi(\mathbf{W}_\ell) \phi_l^{\ell+1}\big)$.

### 4.3 Optimization

Let $\mathcal{L}$ denote the global loss in Eq. 3, the update of $\{\mathbf{W}^\ell\}_\ell$ is achieved using the gradient of $\mathcal{L}$ obtained by *simultaneously* backpropagating the gradients through the networks $g_\theta$, $\phi_r$ and $\phi_l$. More precisely, considering Eq. 3 and $\phi_r^\ell$, $\phi_l^\ell$, the gradient of the global loss w.r.t. $\mathbf{W}^\ell$ is obtained as

$$\frac{\partial\mathcal{L}}{\partial\mathbf{W}^\ell} + \sum_{k=\ell+1}^{L}\frac{\partial\mathcal{L}}{\partial\phi_r^k}\frac{\phi_r^k}{\phi_r^{k-1}}\cdots\frac{\partial\phi_r^{\ell+1}}{\partial\mathbf{W}^\ell} + \sum_{k=1}^{\ell}\frac{\partial\mathcal{L}}{\partial\phi_l^k}\frac{\phi_l^k}{\phi_l^{k+1}}\cdots\frac{\partial\phi_l^\ell}{\partial\mathbf{W}^\ell}, \tag{4}$$

here the left-hand side term in Eq. 4 is obtained by backpropagating the gradient of $\mathcal{L}$ from the output to the input of the network $g_\theta$ whereas the mid terms are obtained by backpropagating the gradients of $\mathcal{L}$ from different layers to the input of $\phi_r$. In contrast, the right-hand side terms are obtained by backpropagating the gradients of $\mathcal{L}$ through $\phi_l$ in the opposite direction. Note that the evaluation of the gradients in Eq. 4 relies on the straight through estimator (STE) [69]; the sigmoid is used as a differentiable surrogate of $h$ during backpropagation while the initial Heaviside is kept when evaluating the responses of $\phi_r$, $\phi_l$ (i.e., forward steps). STE allows training differentiable accessibility networks while guaranteeing binary responses when evaluating these networks.

## 5 Experiments

We evaluate our different GCN architectures on the task of action recognition [71, 28, 20] using the challenging First-Person Hand Action (FPHA) dataset [2]. This dataset, naturally suitable for GCNs, consists of 1175 skeletons whose ground-truth includes 45 action categories with a high variability in style, speed and scale as well as viewpoints. Each video, as a sequence of skeletons, is modeled with a graph $\mathcal{G} = (\mathcal{V}, \mathcal{E})$ whose given node $v_j \in \mathcal{V}$ corresponds to the $j$-th hand-joint trajectory (denoted as $\{\hat{p}_j^t\}_t$) and edge $(v_j, v_i) \in \mathcal{E}$ exists iff the $j$-th and the $i$-th trajectories are spatially neighbors. Each trajectory in $\mathcal{G}$ is described using *temporal chunking* [65, 10]: this is obtained by first splitting the total duration of a video sequence into $M$ equally-sized temporal chunks ($M = 32$ in practice), and assigning trajectory coordinates $\{\hat{p}_j^t\}_t$ to the $M$ chunks (depending on their time stamps), and then concatenating the averages of these chunks in order to produce the raw description (signal) of $v_j$.

**Implementation details and baseline GCN.** Our GCNs are trained end-to-end using Adam [1] for 2,700 epochs with a momentum of 0.9, batch size of 600 and a global learning rate (denoted as $\nu(t)$) set depending on the change of the loss in Eq. 3; when the latter increases (resp. decreases), $\nu(t)$ decreases as $\nu(t) \leftarrow \nu(t-1) \times 0.99$ (resp. increases as $\nu(t) \leftarrow \nu(t-1)/0.99$). The mixing parameter $\eta$ in Eq. 3 is set to 1 and $\lambda$ is slightly overestimated to 10 in order to guarantee the implementation of the targeted pruning rates. All these experiments are run on a GeForce GTX 1070 GPU (with 8 GB memory) and classification performances — as average accuracy through action classes — are evaluated using the protocol in [2] with 600 action sequences for training and 575 for testing. The architecture of our baseline GCN (taken from [65]) consists of an attention layer of 16 heads applied to skeleton graphs whose nodes are encoded with 32-channels, followed by a convolutional layer of 128 filters, and a dense fully connected layer. This initial network architecture is relatively heavy (for a GCN); its includes 2 million parameters and it is accurate compared to the related work on the FPHA benchmark, as shown in Table 1-left. Considering this GCN baseline architecture, our goal is to make it lightweight while maintaining its high accuracy as much as possible.

| Method | Color | Depth | Pose | Accuracy (%) |
|---|---|---|---|---|
| Two stream-color [4] | ✓ | ✗ | ✗ | 61.56 |
| Two stream-flow [4] | ✓ | ✗ | ✗ | 69.91 |
| Two stream-all [4] | ✓ | ✗ | ✗ | 75.30 |
| HOG2-depth [5] | ✗ | ✓ | ✗ | 59.83 |
| HOG2-depth+pose [5] | ✗ | ✓ | ✓ | 66.78 |
| HON4D [7] | ✗ | ✓ | ✗ | 70.61 |
| Novel View [8] | ✗ | ✓ | ✗ | 69.21 |
| 1-layer LSTM [9] | ✗ | ✗ | ✓ | 78.73 |
| 2-layer LSTM [9] | ✗ | ✗ | ✓ | 80.14 |
| Moving Pose [13] | ✗ | ✗ | ✓ | 56.34 |
| Lie Group [14] | ✗ | ✗ | ✓ | 82.69 |
| HBRNN [15] | ✗ | ✗ | ✓ | 77.40 |
| Gram Matrix [16] | ✗ | ✗ | ✓ | 85.39 |
| TF [17] | ✗ | ✗ | ✓ | 80.69 |
| JOULE-color [18] | ✓ | ✗ | ✗ | 66.78 |
| JOULE-depth [18] | ✗ | ✓ | ✗ | 60.17 |
| JOULE-pose [18] | ✗ | ✗ | ✓ | 74.60 |
| JOULE-all [18] | ✓ | ✓ | ✓ | 78.78 |
| Huang et al. [19] | ✗ | ✗ | ✓ | 84.35 |
| Huang et al. [21] | ✗ | ✗ | ✓ | 77.57 |
| Our GCN baseline | ✗ | ✗ | ✓ | **86.08** |

| Pruning rates | TC | # parameters | % of A-C | Accuracy (%) | Observation |
|---|---|---|---|---|---|
| 0% | NA | 1967616 | 100 | 86.08 | Baseline GCN |
| 50.00% | ✗ | 983808 | 100.0 | 86.08 | MP |
| 50.00% | ✓ | 983808 | 100.0 | 86.08 | TCMP (greedy) |
| 49.99% | ✓ | 983836 | 100.0 | 84.34 | TCMP (our) |
| 75.00% | ✗ | 491904 | 99.40 | 85.73 | MP |
| 75.00% | ✓ | 491904 | 100.0 | 85.91 | TCMP (greedy) |
| 75.19% | ✓ | 487990 | 100.0 | 85.21 | TCMP (our) |
| 95.00% | ✗ | 98379 | 72.30 | 83.82 | MP |
| 95.00% | ✓ | 98379 | 100.0 | 84.86 | TCMP (greedy) |
| 95.45% | ✓ | 89453 | 100.0 | **85.21** | TCMP (our) |
| 99.00% | ✗ | 19674 | 21.20 | 76.00 | MP |
| 99.00% | ✓ | 19674 | 100.0 | 80.69 | TCMP (greedy) |
| 99.01% | ✓ | 19285 | 100.0 | **82.95** | TCMP (our) |

Table 1: (Left) Comparison of our baseline GCN architecture against related work on FPHA. (Right) This table shows an ablation study, without TC (i.e., MP) and with TC (i.e., TCMP), for different pruning rates on FPHA. We can see how MP, without TC, produces disconnected network architectures for high pruning rates, and this degrades performances, while TCMP guarantees both Accessibility and Co-Accessibility and also better generalization. Our ablation and comparison are achieved against MP (without TC) [70] and the greedy method (with TC) in [35]. A-C stands for percentage of Accessible and Co-Accessible connections.

**Lightweight CGNs (Comparison & Ablation).** We study the impact of TCMP on the performances of our lightweight GCNs for different pruning rates. Table. 1-right shows the positive impact of TCMP especially on highly pruned network architectures. This impact is less important (and sometimes negative) with low pruning regimes as the resulting architectures have enough (a large number of) Accessible and Co-accessible (AC) connections, so having a few of these connections neither accessible nor co-accessible, i.e. removed, produces a well known regularization effect [44] that enhances performances. In contrast, with high pruning rates and without Topological Consistency (TC), this leads to over-regularized and very disconnected lightweight architectures that suffer from under-fitting. With TC, both accessibility and co-accessibility are guaranteed even with very high pruning regimes, and this also attenuates under-fitting, and ultimately improves generalization as again shown in table 1-right.

# 6 Conclusion

We introduce in this paper a novel lightweight architecture design based on Topologically Consistent Magnitude Pruning (TCMP). The particularity of TCMP resides in its ability to select subnetworks with *only* accessible and co-accessible connections. The latter make the learned lightweight sub-networks topologically consistent and more accurate particularly at very high pruning regimes. The proposed approach relies on two supervisory networks, that implement accessibility and co-accessibility, which are trained simultaneously with the lightweight networks using a novel loss function. Extensive experiments, involving graph convolutional networks, on the challenging task of skeleton-based recognition show the substantial gain of our method.

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
