# OpenReview forum: "Tiny Graph Convolutional Networks  with Topologically Consistent Magnitude Pruning"
_NeurIPS.cc/2023/Workshop/WANT — WANT@NeurIPS 2023 Poster_

### Official Review · Reviewer_JtFc · 2023-10-21
**Enforcing Connected NN when using Magnitude Pruning for Green AI**

**Confidence:** 4

**Review:**

The paper tackles the problem of compressing graph convolutional networks (GCNs) to make them more efficient for deployment. A new loss term is presented that enforces the connectedness in a neural network architecture when it is heavily pruned to reduce computational costs. Only topologically consistent subnetwork connections are selected. The loss - termed TCMP - Topologically Consistent Magnitude Pruning - appears novel to my knowledge. The paper is generally well written and presented. Experiments on the task of skeleton-based action recognition, show benefits of the method for high pruning rates. The method outperforms standard magnitude pruning, showing the benefit of topological consistency especially at high pruning levels.

Issues:

* Equation number missing on page 4
* The manuscript would benefit from diagrams to show accessibility and co-accessibility for toy example
* Only one dataset and task used for evaluation, the paper would benefit from validation on another dataset and task to confirm the findings
* Gains are sometimes marginal to greedy pruning
* The accessibility and co-accessibility concepts are reasonable but lack any theoretical analysis.
* Additional efficiency metrics like inference time, memory usage could better quantify real-world benefits.
* Testing on more GCN architectures and applications would demonstrate generality.

An accept is recommended for the paper, given the interesting and novel contribution. The authors are recommended to tackle the issues above for the final version.

---

### Official Review · Reviewer_wRNv · 2023-10-24
**Efficient Pruning without Losing Connectivity: A Deep Dive into Topologically Consistent Graph Convolutional Networks**

**Confidence:** 4

**Review:**

The paper addresses a pertinent challenge in pruning. By introducing the Topologically Consistent Magnitude Pruning (TCMP) method, the authors provide a solution to preserve the structural integrity of pruned Graph Convolutional Networks, enhancing their generalization capabilities, especially under high pruning conditions.

Strengths:
- Traditional pruning methods might result in topologically inconsistent network architectures, leading to disconnected networks. The TCMP approach explicitly addresses this issue and guarantees the topological consistency of the pruned networks.

- Higher Pruning Rates without Over-Regularization: networks might become weakly discriminant at high pruning rates. The proposed method improves generalization even at very high pruning rates.

Weaknesses

-The authors validate their method on a single dataset.

- The method introduces the concepts of accessibility and co-accessibility to define topological consistency. However, a rigorous theoretical foundation or explanation with proper references for these concepts is missing in Sect. 4.2.

- Correct typos: line 148 lacks a proper reference to the equation.

---

### Official Review · Reviewer_383Q · 2023-10-25
**Submission 40 reveiw**

**Confidence:** 4

**Review:**

This work proposes a topology-aware pruning method for GCN to avoid disconnection at high pruning rates. Empirical evaluation showcases solid performance gain compared to the vanilla magnitude pruning baseline under such pruning rates.

However, the proposed method is only evaluated on a video recognition dataset with no clear justification. This work would benefit from better justifying/demonstrating why the proposed method (and its topology awareness) is beneficial to this particular domain, or alternatively, consider conducting evaluations on general graph datasets like Cora & Reddit. Relevant topology-aware GNN pruning methods should also be compared and discussed outside the skeleton recognition field.

I recommend borderline rejection, given the vision of the current version of this work is relatively unclear, though I'd encourage the author to complete this exploration.

---

### Meta-Review · Area_Chair_6d2t · 2023-10-26

**Recommendation:** Accept (Poster)
**Confidence:** 4

**Metareview:**

The paper proposes a new topology-aware pruning aimed at graph convolutional networks (GCNs), that results on high compression ratios on the datasets used.

As all reviewers acknowledge the novelty of the proposed method and loss component, I believe the paper is of interest to the community of large, hence recommending for acceptance. The authors should seek addressing the reviewers comments for the final version of the manuscript, including validating their results on some additional datasets.

---

### Decision · Program_Chairs · 2023-10-28

**Decision:**

Accept (Poster)

**Comment:**

We thank the authors for their time and contribution to WANT and we are pleased to share that after the reviewing process the paper has been accepted. Congratulations! We encourage the authors to consider reviewers' feedback for the improvement of the camera-ready version. We hope to see you in person at the workshop and brainstorm on efficient training research together!